# Tunable and Switchable Dual-Wavelength SLM Narrow-Linewidth Fiber Laser with a PMFBG-FP Filter Cascaded by Multi-Ring Cavity

Mingquan Gao [1], Bin Yin [1,2,*], Yanzhi Lv [1], Guofeng Sang [1], Benran Hou [1], Haisu Li [3], Muguang Wang [3] and Songhua Wu [1,2]

[1] College of Information Science and Engineering, Ocean University of China, No. 238 Songling Road, Qingdao 266100, China

[2] Laboratory for Regional Oceanography and Numerical Modeling, Qingdao National Laboratory for Marine Science and Technology, Qingdao 266237, China

[3] Key Laboratory of All Optical Network and Advanced Telecommunication Network of EMC, Institute of Lightwave Technology, Beijing Jiaotong University, Beijing 100044, China

* Correspondence: binyin@ouc.edu.cn

**Abstract:** A single longitudinal mode (SLM) dual-wavelength switchable erbium-doped fiber laser (DW-EDFL) based on polarization-maintaining fiber Bragg grating Fabry–Perot cavity (PMFBG-FP) cascaded multiple sub-ring cavities (MSCs) is proposed. A PMFBG-FP with a narrow-band transmission peak and MSCs was implemented as an optical filter to achieve stable dual-wavelength laser output and guaranteed SLM status. By stretching the PMFBG, a highly stable dual-wavelength tunable output could be achieved with a maximum tuning interval of 0.17 nm. The optical signal-to-noise-ratio (OSNR) at dual-wavelength lasing was higher than 57 dB, and the optimal wavelength and power fluctuations within 0.5 h were 0.01 nm and 0.79 dB, respectively. Meanwhile, the measured linewidths of each wavelength were 1.55 kHz and 1.65 kHz, respectively. The measured polarization states of the two laser wavelengths were linear and orthogonal, with a degree of polarization (DOP) of nearly 100%.

**Keywords:** single longitudinal mode; composite sub-cavity; tunable dual-wavelength; orthogonal linear polarization; narrow linewidth

## 1. Introduction

Multi-wavelength single longitudinal mode (SLM) narrow-linewidth erbium-doped fiber lasers (EDFL), with the characteristics of high stability, a high optical signal-to-noise-ratio (OSNR), narrow linewidth, and convenient compatibility, have huge application potential in some important fields such as multi-parameter optical fiber sensing, coherent detection lidar, and optical communication, amongst others [1–4]. Meanwhile, multi-wavelength channel signals have broad application prospects in wavelength division multiplexing (WDM) systems and in the field of generating high-quality terahertz (THz) signals [5,6].

In order to achieve the SLM and narrow-linewidth lasing characteristics, short-line cavities with a distributed Bragg reflector (DBR) structure [7] and a distributed feedback (DFB) structure [8] are the effective approaches. The cavity's length is inversely proportional to the mode's spacing, so a shorter cavity length can widen the longitudinal mode's spacing, which is an effective method utilizing the SLM. However, the length, doping concentration, and uniformity of the gain medium limit the improvement of its optical efficiency [7], and the mode-hopping phenomenon caused by the spatial hole-burning effect caused by the standing wave of the linear cavity is also inevitable.

Moreover, the introduction of a ring cavity can avoid spatial hole-burning and achieve higher optical efficiency caused by the standing wave effect. The optical fiber microstructure of a narrow-band filter, such as the tapered coupling double microsphere cavity [9], the all-fiber Fabry–Perot microcavity [10], and the Fano resonance-based single cylindrical micro-resonator [11], can confine dense longitudinal mode oscillations and suppress mode-hopping. However, the structured optical fiber filter is sensitive to environmental interference, the operability and flexibility in the system are low, and the repetitive manufacturing costs are high. The configuration of multiple sub-ring cavities (MSCs) is a simple method for SLM operation. In recent years, it has attracted extensive attention in the field of single-frequency lasers. It is usually composed of high-quality coupler loops with a specific length to form a composite filter to achieve a more efficient free spectral range due to the Vernier effect [12]. One method is to cascade multiple optical couplers (OCs) in parallel in the main cavity [13]. The other is based on a composite of two OCs with a specific coupling ratio [14,15]. Another realization method is nesting a single-ring structure in a compound double-ring structure [16,17]. There are high requirements for the precise matching of the length of the fibers' composite cavity and the control of the loss to achieve an effective FSR and narrow-band filter bandwidth.

Further, fiber lasers have been added to wavelength channel selection devices such as superimposed high-birefringence fiber Bragg gratings (SI-HBFBG) [18] and polarization-maintaining sampled fiber Bragg gratings (PM-SFBG) to achieve multi-wavelength output [19]. The nonlinear polarization rotation (NPR) effect [15] and the polarization hole burning (PHB) effect [20] can also be introduced into the system loop as an effective way to balance competition. In addition, various techniques such as the introduction of 2D materials [21], the use of tunable optical bandpass filters (TOBF) [22], and variable strain modulation of fiber Bragg gratings (FBG) [16] are used for wavelength tuning. Among them, the axial strain acting on the wavelength channel selection device has good operability and convenience.

In this study, we presented and verified a dual-wavelength tunable single-polarization erbium-doped fiber laser (DW-EDFL) using polarization-maintaining fiber Bragg grating Fabry–Perot (PMFBG-FP) for band selection combined with MSCs to select longitudinal modes through the Vernier effect. The method can use an optical fiber to directly couple with the main laser system; has strong operability, a low-cost filtering structure, and high repeatability; and the optical filter produced has a wide FSR, high stability, and a narrow passband width. Three pairs of wavelengths with a maximum wavelength interval of 0.17 nm can be tuned by rotating the fixed PMFBG fiber displacement stage. The PHB effect is introduced, and the polarization mismatch loss is balanced by the polarization controller (PC), so that the wavelength can work stably in various states of single and dual wavelengths, and the OSNR is higher than 57 dB. During single-wavelength lasing, the DW-EDFL works at the orthogonal linear polarization, the OSNR is higher than 60 dB, and the variation in the wavelength and power of each wavelength is less than 0.01 nm and 0.78 dB, respectively. In addition, the combined filtering effect of the narrow-band transmission peaks of the PMFBG-FP and MSCs effectively suppresses the mode-hopping phenomenon during the long-term operation of the laser system and ensures the stability of single-polarization SLM operation, and the linewidth of each wavelength is less than 1.65 kHz, which is comparable with the current linewidth level of fiber lasers, as few fiber lasers can reach the Hz level [23], and still has certain advantages compared with all-solid-state lasers [24]. By changing the coupling ratio or the combining and matching methods, controlling the ultra-static and stable environment in experiments has great potential for further compressing the linewidth. The system exhibits excellent performance in terms of dual-wavelength tunability and switchability, high stability, SLM, and orthogonal linear polarization.

## 2. Operating Structure and Experimental Principle

### 2.1. DW-EDFL Experimental Configuration

A schematic diagram of the experimental structure for the designed and tested DW-EDFL system is shown in Figure 1. A 980 nm laser diode was used as a pump source to inject into the main ring cavity of the system through a 980/1550 nm wavelength division multiplexer (WDM). The gain medium was a 3 m length of erbium-doped fiber (EDF, YOFC-ED1016, absorption peak at 1532 nm 36 dB/m). The main cavity contained a polarization controller (PC). Through changes in the angular position of the paddles and the polarization state of the light in the laser cavity, the purpose of balancing the gain and loss were achieved. The PMFBG fixed on the fiber displacement stage was connected to the 2-port of the optical circulator which reflected the dual-band channel lasing light into the ring cavity for wavelength primary selection. Through the application of axial strain to the PMFBG, the tunability of the dual-wavelength output was realized. To ensure the unidirectional transmission of the laser light, an optical circulator and an optical isolator (ISO) were adopted.

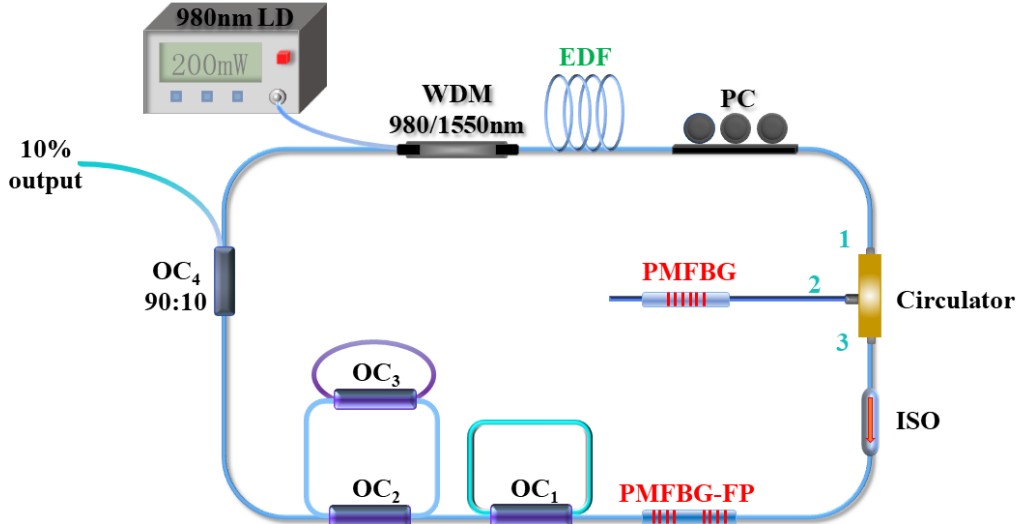

**Figure 1.** Experimental structure diagram of the proposed DW-EDFL.

The main ring cavity length was set to 15 m, which corresponded to the longitudinal mode spacing of 13.8 MHz. The MSCs consisted of three 2 × 2 OCs. $OC_2$ and $OC_3$ formed an "8"-shaped cavity structure in series with $OC_1$. The MSCs had a wide free spectrum range (FSR) through the Vernier effect and acted as a mode filter. Furthermore, we adopted the PMFBG-FP to work together with the multi-ring cavity to achieve a stable single SLM output. The oscillating laser was finally extracted by the 10% port of $OC_4$ and was monitored and measured by an optical spectrum analyzer (OSA, ANDO AQ6317B, 0.01 nm resolution) and an electrical spectrum analyzer (ESA, KEYSIGHT N9020A, 10 Hz–26.5 GHz).

### 2.2. The Principle of Longitudinal Mode Selection

The narrow-band transmission peaks of the PMFBG-FP combined with the MSCs worked together to filter the dense longitudinal modes in the main cavity, ensuring the SLM oscillation. The reflection spectrum of the PMFBG and the transmission spectrum of the PMFBG-FP in the single polarization state of X and Y are shown in Figure 2. The 3 dB bandwidth of the PMFBG reflection peak was 0.22 nm, and the PMFBG-FP has three transmission peaks, which correspond to the tunable laser's output.

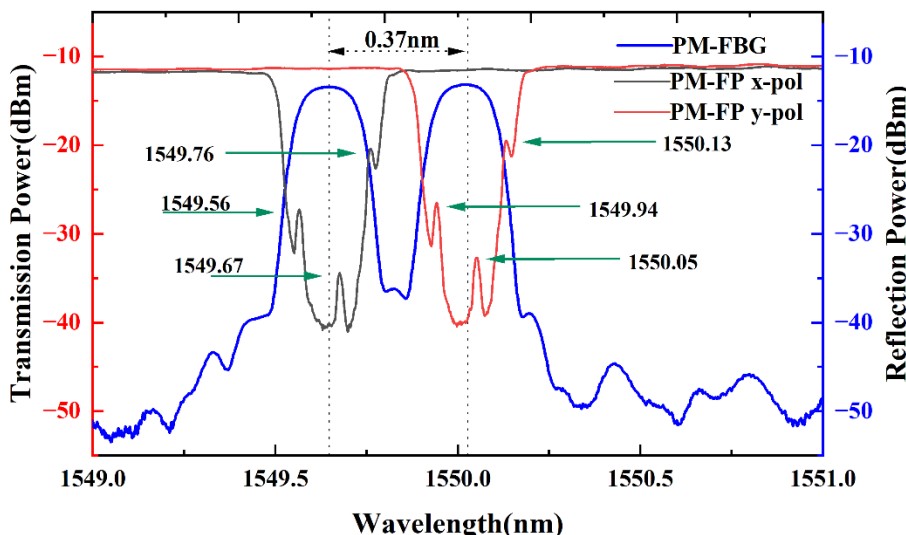

**Figure 2.** Reflection spectrum of the PMFBG (blue curve) and transmission spectrum of the PMFBG-FP in a single polarization state (black and red curves).

To discuss and verify the MSCs' properties individually, the transmission schematic of the MSCs' structure is shown in Figure 3. The MSCs' filter is composed of three $2 \times 2$ OC cascades, where the numbers 1 to 12 represent the port number of each coupler and $E_1$–$E_{12}$ represent the electric field amplitudes of each port. First of all, in $R_1$, the light field at each port can be expressed as [18,25]

$$\begin{pmatrix} E_3 \\ E_4 \end{pmatrix} = \sqrt{1-\gamma_1} \begin{pmatrix} i\sqrt{\alpha_1} & \sqrt{1-\alpha_1} \\ \sqrt{1-\alpha_1} & i\sqrt{\alpha_1} \end{pmatrix} \begin{pmatrix} E_1 \\ E_2 \end{pmatrix} \tag{1}$$

$$E_2 = E_4 \cdot \sqrt{1-\delta} e^{i\omega\tau_1} e^{-\beta L_1} \tag{2}$$

where $\tau_i$ (i = 1,2,3,4) corresponds to the fiber delay at the marked fiber lengths $L_1$–$L_4$, $\omega$ is the angular frequency of the light field, $\delta$ is the splice loss, $\gamma_i$ (i = 1, 2, 3) corresponds to the insertion loss of $OC_1$–$OC_3$, $\alpha_i$ (i = 1, 2, 3) is the coupling ratio of the current OC, and $\beta$ is the fiber loss coefficient. The 4-port light field can be expressed as the following formula:

$$E_4 = \sqrt{1-\gamma_1}(i\sqrt{\alpha_1}E_1 + \sqrt{1-\alpha_1}E_2) \tag{3}$$

According to (1) (2) (3), the transmission formula on the main road is:

$$T = \frac{|E_3|^2}{|E_1|^2} = \sqrt{1-\gamma_1} \cdot$$
$$\frac{(1-\alpha_1)+(1-\gamma_1)(1-\delta)e^{-2\beta L_1}-2\sqrt{1-\gamma_1}\sqrt{1-\delta}\sqrt{1-\alpha_1}e^{-\beta L_1}\cos\omega\tau_1}{1+(1-\gamma_1)(1-\delta)(1-\alpha_1)e^{-2\beta L_1}-2\sqrt{1-\gamma_1}\sqrt{1-\delta}\sqrt{1-\alpha_1}e^{-\beta L_1}\cos\omega\tau_1} \tag{4}$$

Similarly, the transmission matrix through each coupler node of the "8"-shaped cavity filter can be expressed as:

$$\begin{pmatrix} E_{11} \\ E_{12} \end{pmatrix} = \sqrt{1-\gamma_2} \begin{pmatrix} i\sqrt{\alpha_2} & \sqrt{1-\alpha_2} \\ \sqrt{1-\alpha_2} & i\sqrt{\alpha_2} \end{pmatrix} \begin{pmatrix} E_9 \\ E_{10} \end{pmatrix} \tag{5}$$

$$E_5 = E_{12} \cdot \sqrt{1-\delta} e^{i\omega\tau_2} e^{-\beta L_2} \tag{6}$$

$$\begin{pmatrix} E_7 \\ E_8 \end{pmatrix} = \sqrt{1-\gamma_3} \begin{pmatrix} i\sqrt{\alpha_3} & \sqrt{1-\alpha_3} \\ \sqrt{1-\alpha_3} & i\sqrt{\alpha_3} \end{pmatrix} \begin{pmatrix} E_5 \\ E_6 \end{pmatrix} \tag{7}$$

$$E_6 = E_8 \cdot \sqrt{1-\delta} e^{i\omega\tau_3} e^{-\beta L_3} \tag{8}$$

$$E_{10} = E_7 \cdot \sqrt{1 - \delta} e^{i\omega\tau_4} e^{-\beta L_4} \tag{9}$$

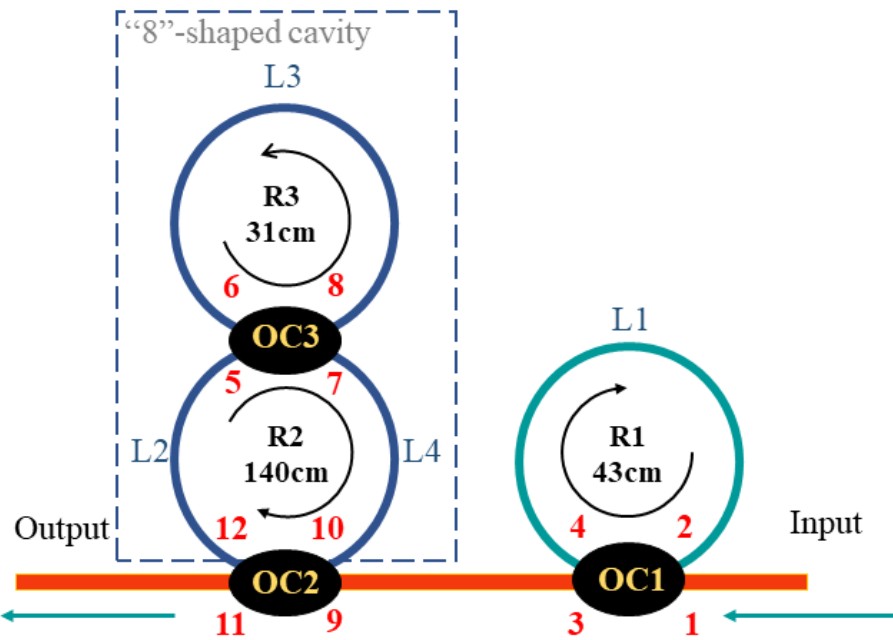

**Figure 3.** Schematic diagram of the three-loop MSCs filter.

The input and output ports were cycled enough times to reach a steady state.

According to the bandwidth of the fiber filter and the length of the main cavity, we set the length of $R_1$ to 43 cm. In addition, the lengths of $R_2$ and $R_3$ in the "8"-shaped cavity were 140 cm and 31 cm, respectively; the corresponding FSRs were 480 MHz, 147 MHz, and 672 MHz, as shown in Figure 4b–d, respectively. Therefore, the effective *FSRc* of the MSCs is expressed as [26]

$$FSR_c = q_1 \frac{c}{nL_1} = q_2 \frac{c}{nL_2} = q_3 \frac{c}{nL_3} \tag{10}$$

where n is the core refractive index, *L* is the length of the ring cavity, and the integer *q* is the longitudinal mode coefficient. The effective *FSRc* of MSCs was 23.5 GHz, and the PMFBG reflection bandwidth could not meet the requirements of stable SLM. Through the combined action of the narrow-band PMFBG-FP, stable single longitudinal mode output could be achieved. The three-ring structure avoided the experimental inaccuracy caused by the calculation error and the actual fiber cutting and fusion error in the context of guaranteeing the above conditions.

In addition, according to the role of MSCs in expanding the longitudinal mode spacing, it must be ensured that the 3 dB passband of MSCs is 0.5–1 times the interval between adjacent longitudinal modes of the main cavity [18]. The 3 dB bandwidth of $R_2$ can be calculated as 27.5 MHz according to the following formula

$$\Delta\nu = \frac{1}{2\pi\tau_c} \tag{11}$$

where $\tau_c$ is the photon lifetime in the resonant loop, which can be described as

$$\tau_c = \frac{nL}{c\sigma} \tag{12}$$

where *L* is the length of the passive resonant loop, c is the speed of light in a vacuum, n ≈ 1.47 is the refractive index of the fiber core, and $\sigma$ is the one-way transmission loss

in $R_2$, which is mainly determined by thoupling ratio of the optical coupler and can be expressed by the following equation

$$\sigma = \ln\left(\frac{I_0}{I_1}\right) \tag{13}$$

where $I_0$ and $I_1$ represent the light intensity of the input light and the output light in $R_2$, respectively. However, when we actually measured the coupling ratio of the coupler with a power meter, we found that the coupling ratio was slightly larger than 0.5 and close to 0.56.

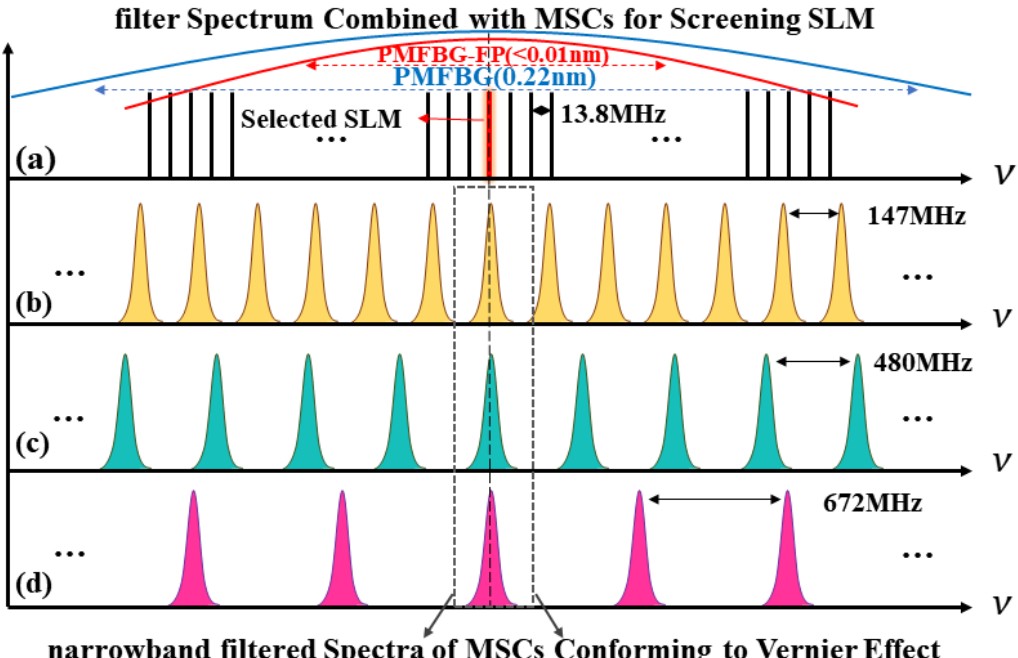

**Figure 4.** Schematic diagram of efficient FSR selection of SLM based on the Vernier effect. (**a**) Longitudinal mode spacing of the main cavity covered in the PMFBG and PMFBG-FP filtered main peaks and the selected SLM; (**b–d**) are the *FSRs* of $R_2$, $R_1$, and $R_3$, respectively.

According to the data parameters of the provided filter and the main cavity length, MSCs can choose one of the nearly 200 modes of oscillation.

## 3. Experiment Results and Discussions

### 3.1. Spectral Characteristics of the Proposed Laser

The experiments were performed on an optical table in a clean laboratory with an ambient temperature of 26 °C, and the fused loops of the MSCs were fixed on a vibration isolation plate. Under the condition of 400 mW of pump power (above the pump threshold of 120 mW), the dual-wavelength output and the single-wavelength switchable output could be realized by changing the angle of the PC, which was measured by an ANDO AQ6317B OSA with a resolution of 0.01 nm, as shown in Figure 5a. The two center wavelengths $\lambda_1$ and $\lambda_2$ were at 1549.66 nm and 1550.03 nm, with a 3 dB bandwidth of 0.08 nm and an OSNR of more than 57 dB; the wavelength separation was 0.37 nm. Furthermore, we verified the wavelength and power stability of the DW-EDFL at room temperature at two-minute intervals within half an hour, as shown in Figure 5b,c. As can be seen, the maximum fluctuation in the wavelength and power at $\lambda_1$ and $\lambda_2$ were 0.01 nm and 0.79 dB, and 0.01 nm and 1.13 dB, respectively.

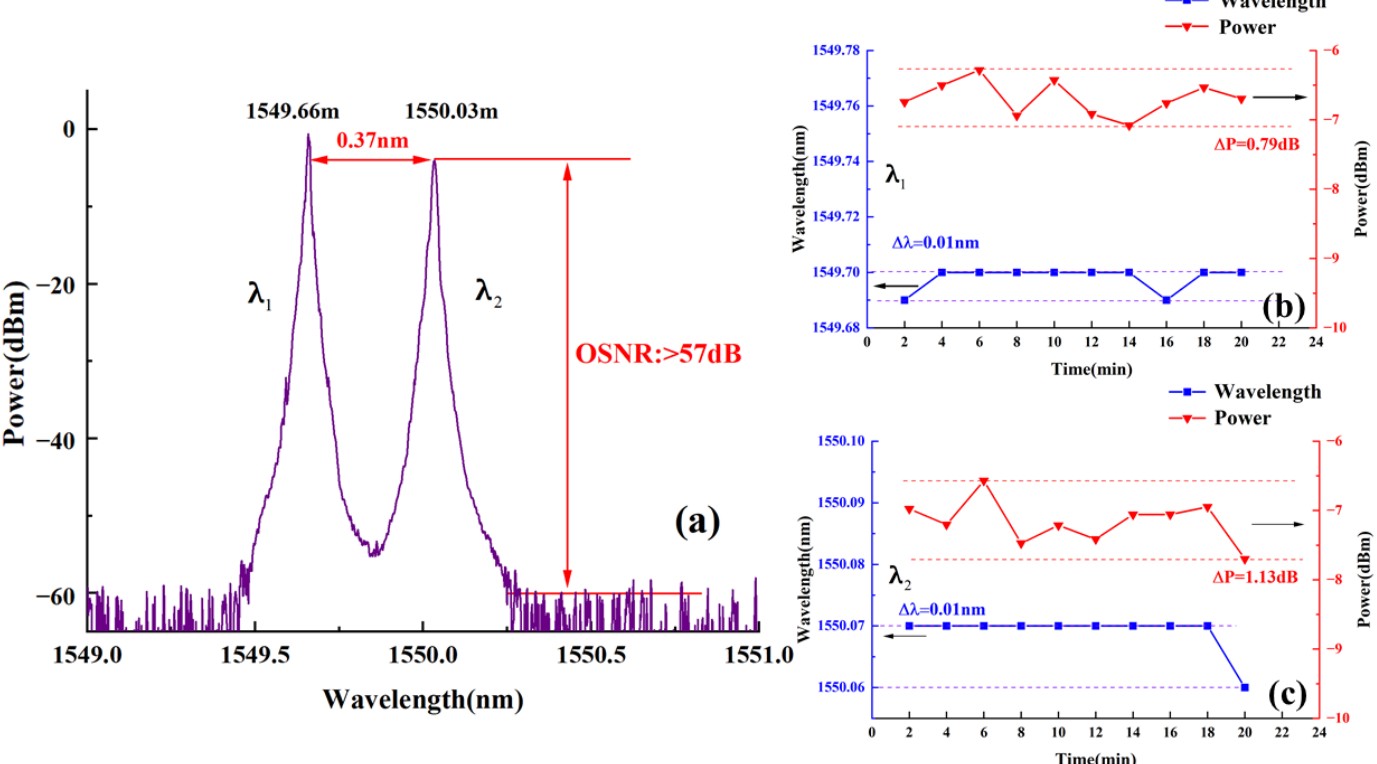

**Figure 5.** (**a**) Dual wavelength operating spectrum measured by the OSA at 400 mW pump power; (**b**,**c**) the variation in the lasing output's power intensity and the wavelength during 0.5 h.

Next, the gain competition caused by simultaneous dual-wavelength lasing and the influence of external environment may affect the output laser's performance. We further investigated the performance after switching to single-wavelength lasing. At a pump power of 400 mW, by carefully tuning the PC, strong gain competition was introduced at different wavelengths, resulting in stable single-wavelength lasing with OSNR > 60 dB for each wavelength, as shown in Figure 6a,c, illustrating good beam quality. The optical spectrum was scanned 10 times with a 2-min interval between each scan. The experimental results in Figure 6b,d show that the single-wavelength and fluctuations in power at $\lambda_1$ and $\lambda_2$ were 0.01 nm with 0.78 dB, and 0.01 nm with 0.23 dB, respectively.

Additionally, when the axial strain was applied to the PMFBG fixed on the optical fiber's displacement platform, its central wavelength (with a 3 dB bandwidth of 0.22 nm) drifted from about 1549 nm to 1551 nm, ensuring that the central wavelength of the transmission channel coincided with the central wavelength of the reflection channel during the tuning process of applying stress, which is an advantageous position in the gain competition. As shown in Figure 7, the dual-wavelength interval was maintained between 0.36 and 0.37 nm, and the tunable range was 0.17 nm, which basically corresponded to the interval range of the narrow-band transmission peak.

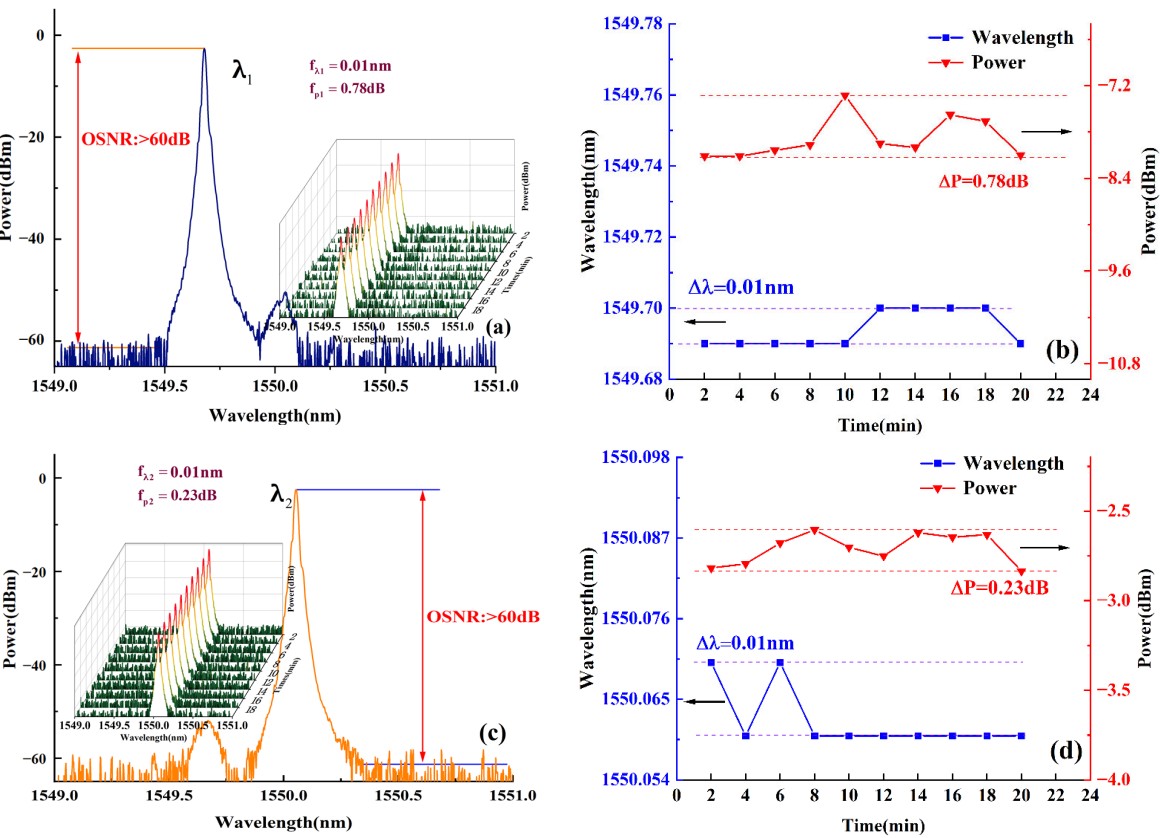

**Figure 6.** Optical spectrum and stability performance of single-wavelength lasing. (**a**,**b**) Typical spectrum and wavelength stability at $\lambda_1$; (**c**,**d**) typical spectrum and wavelength stability at $\lambda_2$.

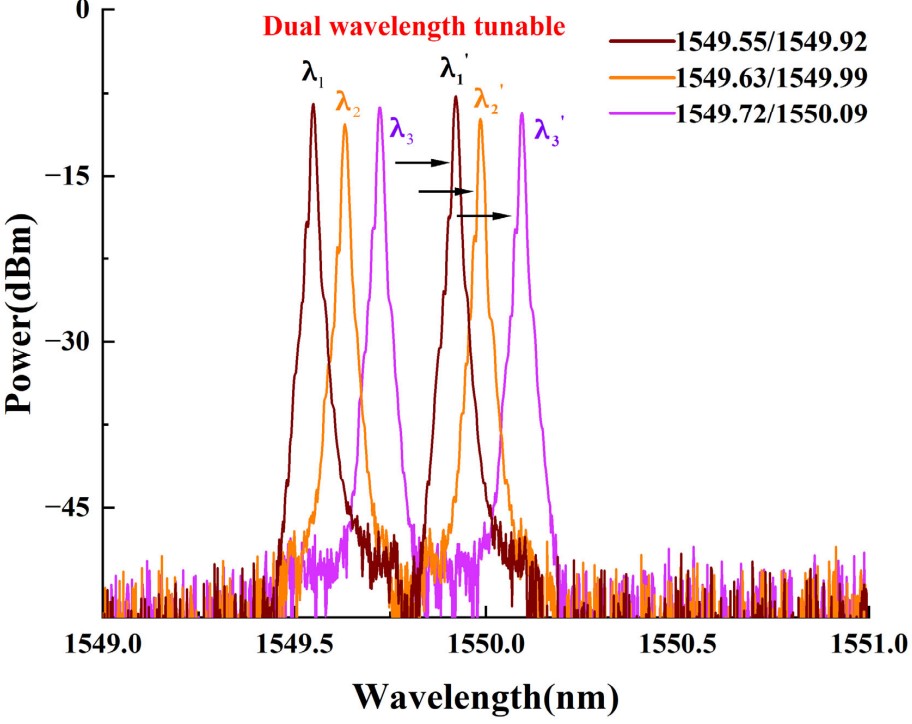

**Figure 7.** Tunability of dual-wavelength operation, where 1549.92, 1549.99, and 1550.09 nm correspond to the tuned wavelengths of 1549.55, 1549.63, and 1549.72 nm respectively.

The relationship between the output power of the DW-EDFL at $\lambda_1$ and the pumping power is shown in Figure 8. The pump threshold of the laser was about 120 mW. When the pumping power was 310 mW, the output power reached 3.48 mW and no power output was saturated. The curve exhibited good linearity ($R^2$~0.993), with a fitted slope efficiency of 1.8%. this shows that the output power of the laser can continue to increase with an increase in the pumping power. Since the mode field mismatch may occur in different types of fiber splices in the laser's cavity, considering the insertion loss at the flange interface and the splicing loss caused by multiple splices in the MSCs' structure, the laser's efficiency may be affected.

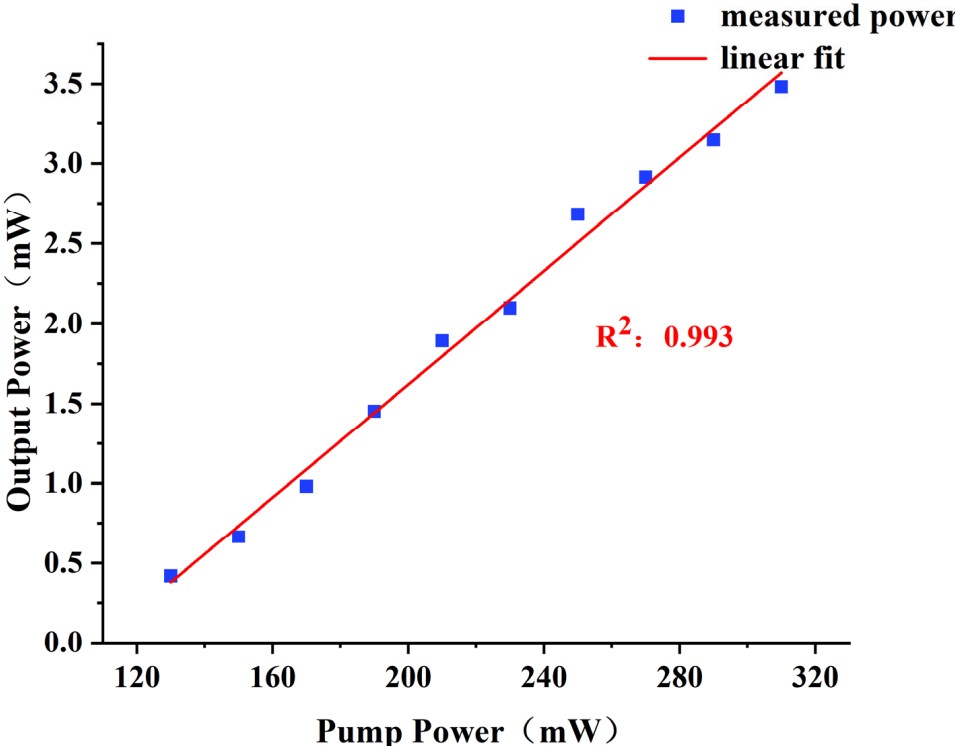

**Figure 8.** Corresponding relationship between pumping power and the output power of the DW-EDFL.

### 3.2. Verification of the SLM

In order to verify the SLM performance of the DW-EDFL, the self-homodyne method was exploited here. The output laser was connected to a 200 MHz photodetector (PD), and the PD was connected to the ESA through a radio frequency (RF) line to monitor its beat frequency signal. The spectrum scanning range of the ESA was 0–200 MHz, and the resolution bandwidth (RBW) at 100 kHz was set to 0.5 h. Figure 9a,b shows the output state of the proposed laser during single-wavelength operation based on PMFBF-FP cascaded MSCs. It was found that no beat frequency signal was captured, indicating that the DW-EDFL was in a stable SLM operation state. As show in Figure 9c, only the MSCs were retained in the ring cavity, and without the limitation of the PMFBG-FP narrow-band filter channel, mode-hopping occurred during the observation time. When the MSCs were removed, the ESA scanned obvious beat signals, mainly in the low-frequency range, indicating that there was a MLM state, the details of which are shown in Figure 9d. As shown in Figure 9e, the "8"-shaped cavity and the PMFBG remained in the main ring cavity, and the results showed that there were several beat signals, indicating the existence of the MLM phenomenon. However, as shown in Figure 9f, when we cascaded the "8"-shaped cavity with the PMFBG-FP, the signals captured during the accumulation time may have had mode-hopping caused by environmental influences or internal structural errors in the system. Each filter plays a certain role in the filtering effect, but to maintain the operation of the SLM, a cascade combination is needed to achieve this effect.

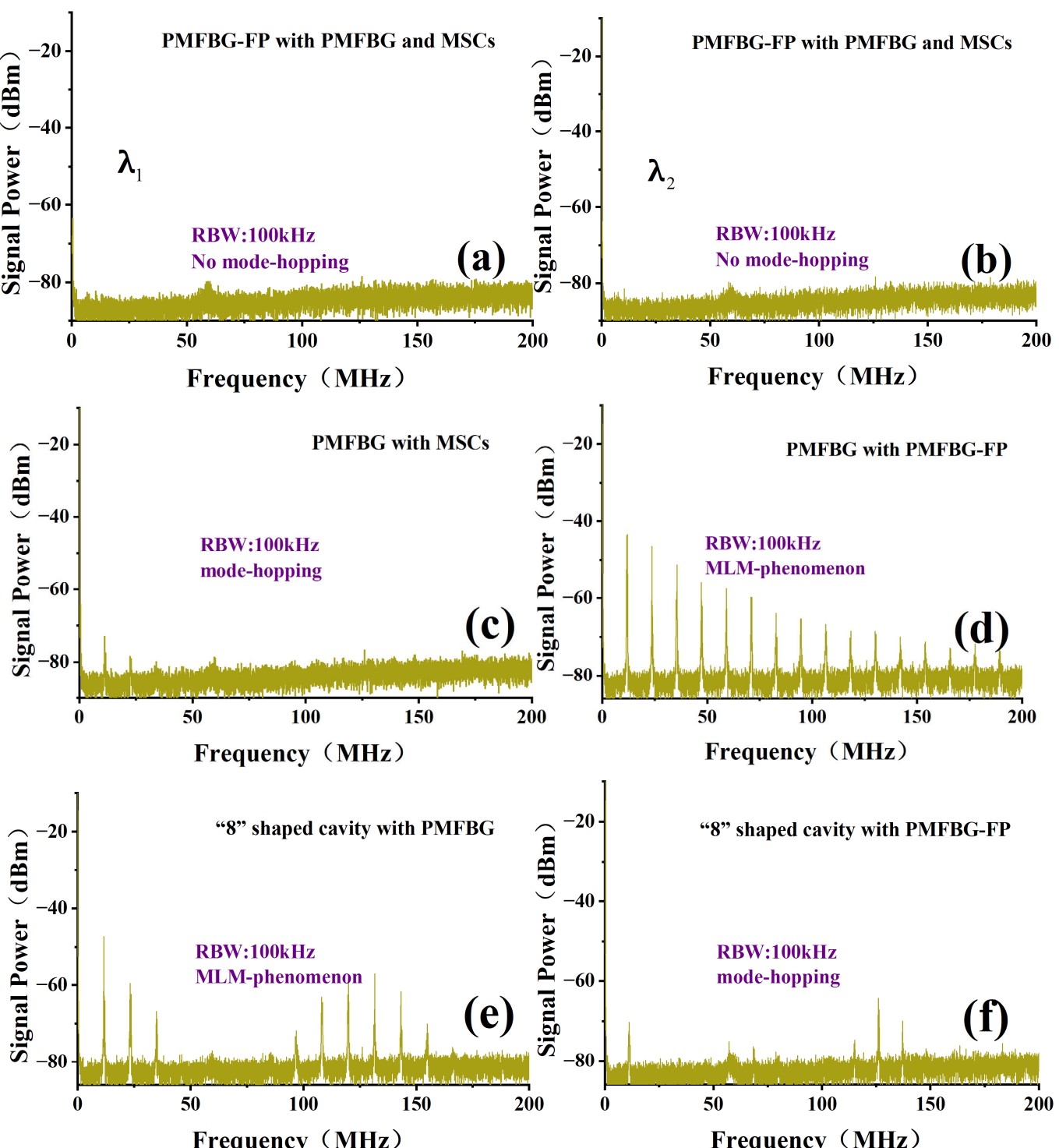

**Figure 9.** Electrical spectra of the beat signal accumulated within half an hour. (**a,b**) Beat signals of $\lambda_1$ and $\lambda_2$ under the conditions of a combination of narrow-band PMFBG-FPs with PMFBG and MSCs, respectively. (**c**) Only MSCs were retained without binding the PMFBG-FP. (**d**) Only the PMFBG-FP was retained without binding the MSCs. (**e**) Only the "8"-shaped cavity is retained without binding the PMFBG-FP. (**f**) The "8"-shaped cavity cascade PMFBG-FP.

### 3.3. The Measured Linewidth Characteristics

The linewidth of each wavelength of the DW-EDFL was measured using the delayed self-heterodyne interferometer (DSHI) method [27]. In order to ensure the accuracy of the

linewidth's measurements and obtain a good Lorentzian line shape of the photocurrent spectrum, the system used a 40 km delay fiber, corresponding to a nominal resolution of 4.9 kHz. The 100 MHz acousto-optic modulators (AOMs) were used for frequency shifting, which eliminated the zero frequency interference. Figure 10a,b shows the frequency spectrum (the black curve) and the fitted Lorentzian curves (the red curve) of the $\lambda_1$ and $\lambda_2$ wavelengths of the dual-wavelength laser at 100 Hz RBW, respectively. Calculation of the 20 dB bandwidth of the fitting curve showed that the linewidths of the two switchable single-wavelength lasers were 1.55 kHz and 1.65 kHz, respectively, for $\lambda_1$ and $\lambda_2$. It is worth noting that due to the limitation of the length of the delay fiber and the unavoidable influence of the current noise of the output laser, the measurement resulted in a broadening of the self-interference spectrum, and the actual linewidth was smaller than the measured value.

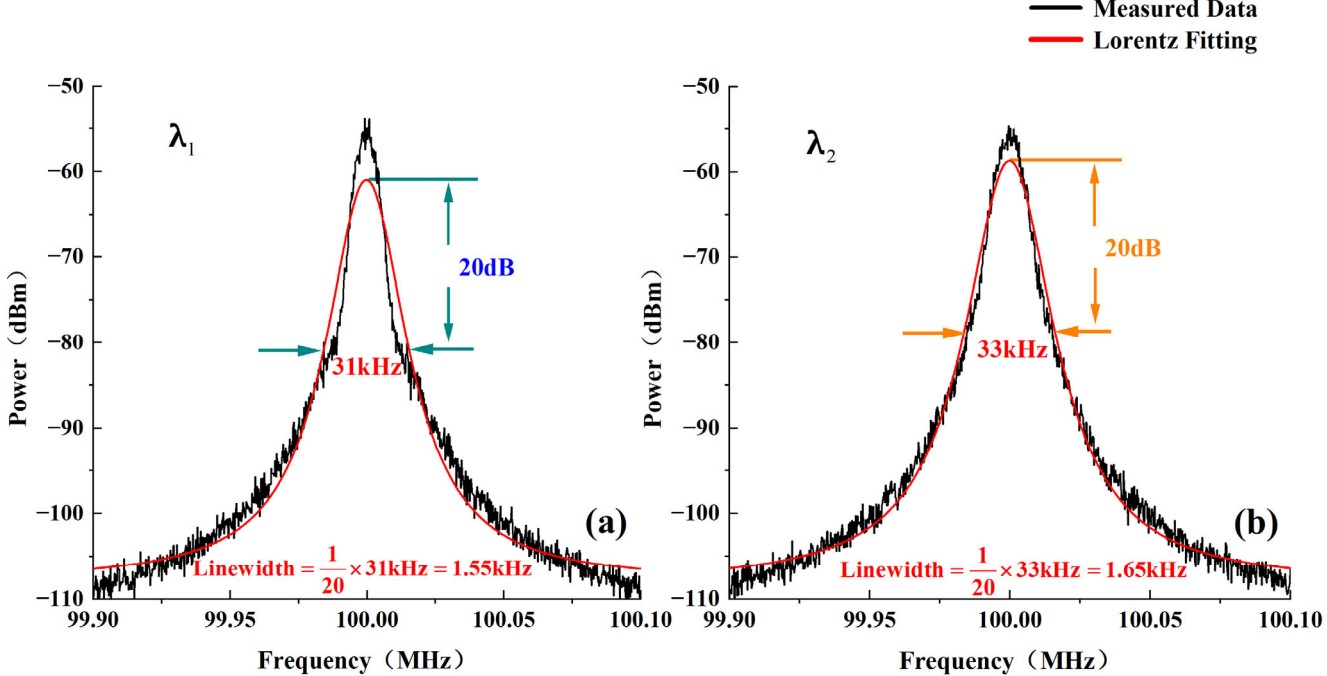

**Figure 10.** Measurement of the laser's linewidth by DSHI: measurement results for (**a**) $\lambda_1$ and (**b**) $\lambda_2$.

### 3.4. The Measured Polarization Characteristics

Finally, we measured and analyzed the state of polarization (SOP) characteristics of the proposed laser at the $\lambda_1$ and $\lambda_2$ wavelength outputs using a polarization analyzer (Thorlabs, PAX1000IR2/M) [28], as shown in Figure 11a,b. During the measurement process, the characteristics were accumulated for 5 min in a stable environment. It can be seen that the SOP of the two wavelengths is nearly orthogonal. The dot trace on the Poincare sphere shows only a fixed point and no drift trace, which indicate that the output laser was in a stable single polarization state. The degree of polarization (DOP) of the fiber laser was studied further, and the polarization degrees of the dual wavelengths $\lambda_1$ and $\lambda_2$ reached 96.04% and 101.91%, respectively (the DOP expressed as a percentage here should be less than 100%, as the analytical error of the equipment leads to deviation in the measurements). The results showed that the DW-EDFL exhibited a stable linear polarization state.

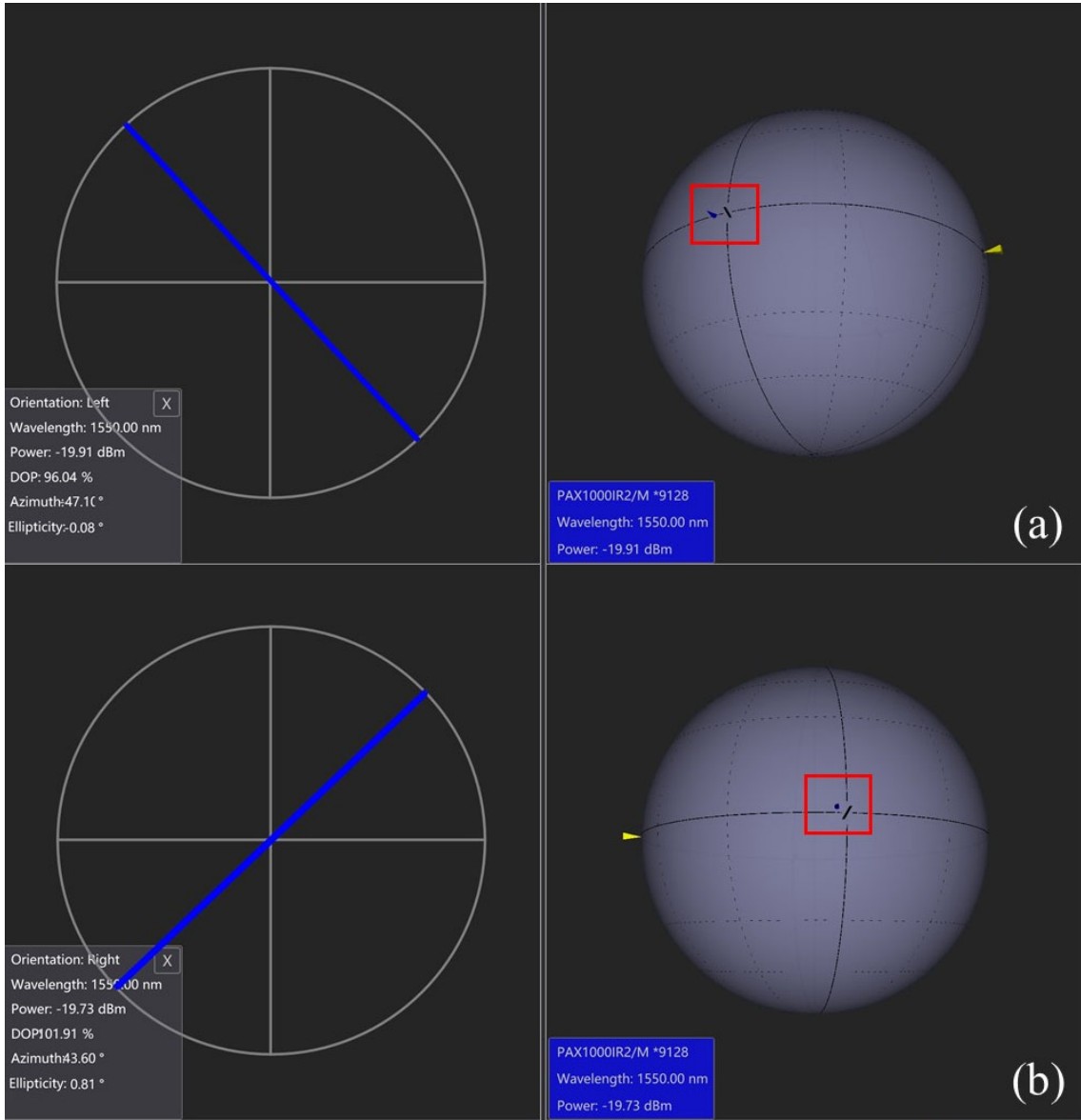

**Figure 11.** Polarization parameters of the fiber lasers at the (**a**) $\lambda_1$ and (**b**) $\lambda_2$ wavelengths.

## 4. Conclusions

An efficient and stable dual-wavelength SLM tunable EDFL of MSCs-cascaded PMFBG-FP narrow-band filters with a tunable range of 0.17 nm was proposed and demonstrated. We introduced three-ring MSCs to effectively avoid the calculation error and experimental welding error of the cavity's length according to the Vernier principle for mode selection. The introduction of the PHB effect provided conditions for stable simultaneous dual-wavelength output, so that the OSNR was higher than 57 dB, and the wavelength and fluctuations in the power of the dual-wavelength laser were less than 0.01 nm and 0.79 dB, and 0.01 nm and 1.13 dB, respectively. During the single-wavelength working stage, the OSNR in the measurement within 30 min was higher than 60 dB, and the maximum wavelength and fluctuations in power were 0.01 nm and 0.78 dB, respectively. In order to verify the SLM's characteristics, the beat frequency signals of the laser output were observed under different combinations of MSCs or under different cascade conditions. Each optical device played an irreplaceable role in the selection of the mode and the realization of the SLM's output. The linewidth of the $\lambda_1$ and $\lambda_2$ wavelength outputs could reach 1.55 kHz and 1.65 kHz. For each wavelength's operation state, the two wavelengths had linearly

orthogonal SOPs and stable DOPs within the monitoring time. The proposed SLM tunable and switchable DW-EDFL may have potential applications in the field of long-distance multi-parameter optical fiber sensing and coherent lidar detection.

## 5. Disclosures

The authors declare that they have no known competing financial interests or personal relationships that could have appeared to influence the work reported for this study.

**Author Contributions:** Conceptualization, B.Y. and M.G.; methodology, B.Y. and M.G.; validation, M.G. and Y.L.; investigation, H.L.; resources, M.W.; writing—original draft preparation, M.G.; writing—review and editing, B.Y. and M.G.; visualization, G.S. and B.H.; supervision, S.W.; All authors have read and agreed to the published version of the manuscript.

**Funding:** This research was funded by National Natural Science Foundation of China (61901429, U2006217, 62075007) and the Natural Science Foundation of Shandong Province (ZR2019BF003), the Fundamental Research Funds for the Central Universities (202265004).

**Acknowledgments:** This work was supported by National Natural Science Foundation of China (61901429, U2006217, 62075007) and the Natural Science Foundation of Shandong Province (ZR2019BF003), the Fundamental Research Funds for the Central Universities (202265004).

**Conflicts of Interest:** The authors declare no conflict of interest.

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
