# Peer review of "Tunable and Switchable Dual-Wavelength SLM Narrow-Linewidth Fiber Laser with a PMFBG-FP Filter Cascaded by Multi-Ring Cavity"

_photonics, doi:10.3390/photonics9100756_

Round 1

Reviewer 1 Report

This manuscript describes a dual-wavelength switchable erbium-doped fiber laser. The single-longitudinal-mode operation, the long-term stability (0.79-dB power variation in 0.5 h), the narrow linewidths (e.g., about 1.55 kHz), and the linear polarization of the laser output make the system attractive for many applications, including THz wave generation and optical sensing. The proposed laser configuration is well characterized. Overall, the manuscript is publishable. But the quality of this paper could be improved. Also, there are several issues regarding the manuscript that should be addressed.

1) The introduction part can be improved. For instance, one or more specific challenges the manuscript targeted should be clearly identified. Although previous works have been summarized, the difficulties they faced are unclear. Also, how the proposed method overcomes these difficulties and benefits the field of multi-wavelength lasers is not presented. Achieving kHz-level linewidths for a multi-wavelength laser is good. But how this valve compares to state-of-the-art systems should be discussed. More importantly, the novelty and importance of this work should be highlighted in the introduction.

 2) Part 3. The spectral resolution of the spectrometer (AQ6317B OSA) is set to 0.01 nm. How come the maximum fluctuation of the wavelength is measured to be 0.002 nm? (Line 187)

 3) Part 3.3. Usually, the linewidth can be fitted with a Lorentzian line shape. But, for Fig. 10, the Lorentzian fitting does not match the measurement data very well. What would be the possible reasons?

4) A few inappropriate phrases should be corrected. For instance, “signal to noise ratio” should be corrected to “signal-to-noise-ratio”, “(a) (b) are the measurement results for l1 and l2.” (Line 271) to “(a) (b) are the measurement results for l1 and l2, respectively.”
